# A Virtual Sensing Concept for Nitrogen and Phosphorus Monitoring Using Machine Learning Techniques

**DOI:** 10.3390/s22197338

**Published:** 2022-09-27

**Authors:** Thulane Paepae, Pitshou N. Bokoro, Kyandoghere Kyamakya

**Affiliations:** 1Department of Electrical and Electronic Engineering Technology, University of Johannesburg, Doornfontein 2028, South Africa; 2Institute for Smart Systems Technologies, Transportation Informatics, Alpen-Adria Universität Klagenfurt, 9020 Klagenfurt, Austria

**Keywords:** water quality monitoring, specification book, baseline model, accuracy benchmark, data scaling, missing values handling, surrogate parameters, soft-sensor, machine learning

## Abstract

Harmful cyanobacterial bloom (HCB) is problematic for drinking water treatment, and some of its strains can produce toxins that significantly affect human health. To better control eutrophication and HCB, catchment managers need to continuously keep track of nitrogen (N) and phosphorus (P) in the water bodies. However, the high-frequency monitoring of these water quality indicators is not economical. In these cases, machine learning techniques may serve as viable alternatives since they can learn directly from the available surrogate data. In the present work, a random forest, extremely randomized trees (ET), extreme gradient boosting, k-nearest neighbors, a light gradient boosting machine, and bagging regressor-based virtual sensors were used to predict N and P in two catchments with contrasting land uses. The effect of data scaling and missing value imputation were also assessed, while the Shapley additive explanations were used to rank feature importance. A specification book, sensitivity analysis, and best practices for developing virtual sensors are discussed. Results show that ET, MinMax scaler, and a multivariate imputer were the best predictive model, scaler, and imputer, respectively. The highest predictive performance, reported in terms of R^2^, was 97% in the rural catchment and 82% in an urban catchment.

## 1. Introduction

### 1.1. Background and Motivation

Eutrophication (Eutro) and harmful cyanobacterial bloom (HCB) are causes of concern around the world, with vital water bodies frequently experiencing increasingly toxic blooms in the past few decades [1]. Eutro refers to the nutrient enrichment of water mainly from agricultural sources, wastewater discharges, industrial wastes, and urban runoff [2], while HCB is one of the most severe symptoms of Eutro [3]. HCB is problematic for the drinking water treatment process [4], and some strains can produce toxins (cyanotoxins) that have significant health effects on humans, animals, and other organisms [5].

In addition, the severity of Eutro and HCB has significant ecological and socio-economic costs due to the (i) increase in costs associated with water treatment, (ii) adverse effects on waterfront real estate, (iii) tourism and recreational losses due to the negative perceptions of water quality (WQ), (iv) animal fatalities, and (v) fish mortality [6]. Africa is particularly vulnerable to these economic losses due to its lag in socio-economic development. This assertion is substantiated by the fact that the combined costs of eutrophication in England and Wales and the USA were about US$160 million per year in 2003 [7] and US$2.2 billion per year in 2009 [8], respectively.

Therefore, water bodies need to maintain an acceptable chemical, physical, and biological status in order to preserve raw water supply, protect human and animal health, and safeguard the natural ecosystems and biodiversity. To achieve this, catchment managers need to continuously keep track of key WQ indicators. Nitrogen (N) and phosphorus (P) are the two major nutrient drivers of surface water Eutro [3]. Thus, efficient control and monitoring of these nutrients, in the context of reducing the risk of Eutro, depends on the cost, accuracy, reliability, and capability of the corresponding measurement process to collect continuous in-situ data in real-time [9,10].

During the pre-industrial era, nutrient (primarily N and P) concentrations in water bodies were generally low [11]; hence infrequent (i.e., weekly or monthly) monitoring through discrete sampling campaigns was sufficient [12]. However, recent increases in municipal and industrial wastewater releases, which in some cases are only partially treated, and the increased production of nutrients to support agricultural fertilization have dramatically increased their presence in water bodies [11]. Consequently, the commonly applied monitoring approach that relies on analyzing grab samples in laboratories is no longer effective, particularly in terms of analysis costs and delays in data acquisition [13]. These drawbacks highlight the need for cost-effective, reliable, and accurate sensors for continuous in-situ monitoring of these nutrients.

Commercially, several sensor technologies are available for measuring nutrient concentrations. These technologies can be classified into nonchemical optical sensors, ion-selective electrodes (ISE), and wet-chemical analyzers. While ISE (commonly used for measuring N) are relatively cheaper, they are inaccurate and susceptible to interferences and significant drifting over shorter periods [14], making their frequent calibration necessary for data reliability. In contrast, optical sensors and wet-chemical analyzers (commonly used for measuring P) show higher resolution, accuracy, and precision, but are more expensive and have high power requirements and maintenance costs [14]. These shortcomings may limit their long-term deployment for continuous (or high-frequency) outdoor monitoring.

In these cases, machine learning (ML) techniques may serve as viable alternatives since they can learn directly from the available WQ data. These techniques can enable the prediction of target variables (e.g., N and P) using other easily and continuously measurable WQ variables as surrogates. The model that exploits available surrogate data to infer the target (or hard-to-measure) variables is called a virtual (or soft) sensor. Figure 1 shows a conceptual representation of the virtual sensing system.

Some key factors that affect the operation of these virtual sensors include the complexity of the model, acceptable predictive accuracy, the type and number of surrogate sensors, and the availability of high-quality training data [9]. Thus, the core issue is to develop robust and inexpensive virtual sensors with excellent performance.

### 1.2. Literature Review

We are aware of only three studies that have assessed the feasibility of ML techniques for predicting nutrient concentrations, particularly for WQ monitoring purposes. Castrillo and García [9] used multiple linear regression (MLR) and random forest (RF) models to estimate nutrient concentrations in the rural and urban catchments of the River Thames in the UK. They considered only the variables most commonly measured in real-time (in-situ) as predictors (or surrogates). However, the modeling accuracy, reported in terms of the root mean squared error (RMSE) and the normalized RMSE (nRMSE), is not state-of-the-art (except for nitrate). For instance, the average nRMSE for both catchments is 21% and increases to 24% when excluding nitrate prediction in the rural catchment.

Elsewhere, Ha et al. [3] also used RF and MLR models to estimate nutrient concentrations in the Tri An reservoir in Vietnam. Amongst the predictors, they included chemical and biochemical oxygen demand. However, high-frequency monitoring of both these variables is not feasible due to their sensors’ reliability and prohibitive cost [15,16]. Although the predictive performance (reported in terms of the RMSE and the coefficient of determination (R^2^)) in the validation phase was tolerable (average R^2^ was 0.858), the results may be biased since the dataset is relatively small, in which case, the results depend on which data was used for model training vs. validation. The performance assessment based on the data split randomly over the entire dataset (as opposed to the year-based split) may have enabled a more robust conclusion.

Recently, Harrison et al. [12] employed RF to estimate nutrient concentrations within a forested area (Lake George) in New York, USA. They mainly used the variables with high-frequency sensor data (except for fluorescent dissolved organic matter and soil moisture) as surrogates. However, the predictive performance, reported in terms of the Nash-Sutcliffe Efficiency (NSE), was not state-of-the-art, especially for phosphorus fractions. For instance, the NSE was 32% for soluble reactive P, 47% for total soluble P, 74% for total P, and 76% for particulate P.

Therefore, despite the recognition of the urgent need for accurate, reliable, and cost-effective monitoring programs [17], scientific progress has been rather sluggish. Furthermore, two of the three studies [3,12] build their predictive models using the data from one watershed of interest. Thus, even though the authors reported the accuracies better than [9] (who used two watersheds with contrasting land uses), the conclusions from their studies are not adequately substantiated since the robustness of their models may be limited [18]. Additionally, apart from applying the same ML algorithm (RF) across different study areas, all studies failed to achieve the target accuracy (defined in Section 3.1.3) for most variables, particularly P fractions. This drawback may impede the uptake of virtual sensing for operational purposes. However, the success of other ensemble techniques such as the extremely randomized trees algorithm [19], which has been tremendous in various application domains in recent years, is yet to be explored for N and P prediction.

In addition, there is also a lack of (i) effective baseline models and (ii) established datasets and accuracy benchmarks. For instance, MLR is not an appropriate baseline since WQ data often exhibit multicollinearity, nonlinear relationships, and skewed distributions [12]. Additionally, each work used different datasets since there is no standard benchmark, and the accuracy is reported using different performance metrics. In the absence of open-source implementations, these inconsistencies make it difficult to compare and critically assess the models’ performance, and this impedes the research progress. Further, it is unclear what predictive accuracy would be desired to trigger management actions when nutrient concentrations exceed the regulatory thresholds. Accordingly, there is a need for a specification book that would form a basis for a robust and rigorous virtual sensor development process. Such a specification book, including the provision of adequate modeling details, would also make the experiments repeatable and results reproducible; the two issues essential for advancing the research field [20].

The other common issue with WQ data is the high number of outliers [21]. For example, except for temperature, all the variables in the present work have outliers (values outside the whiskers of a boxplot). A common procedure is to use either the three-sigma rule [22] or the interquartile range method [23] to identify and delete the outliers. However, because of the multivariate nature and seasonal variability of WQ data [21,24], removing these outliers may not be the most effective procedure since they may be legitimate, and we do not want to lose their impact. Consequently, this directly impacts the method used for data scaling since some scalers are very sensitive to the presence of outliers. Although the performance of metric and gradient-based ML algorithms depends on appropriately scaled data [25], the effect of various scalers and transformers on the predictive performance of N and P concentrations is yet to be assessed.

Another prevalent issue is the presence of many missing values (MVs) in WQ data. For instance, the raw data in the present work (described in Section 2.1) have a missing rate as high as 41% (total reactive P) in one catchment. How these missing data are handled remains crucial since many ML algorithms do not support data with MVs. A commonly applied basic strategy is to discard the entire columns and/or rows containing MVs. For example, Castrillo and García [9] managed the MVs by deleting the rows that contained such values. However, this has the drawbacks of (i) losing data which may be valuable even though incomplete and (ii) reducing the size of the datasets, which on many occasions are already small. For instance, the four-year dataset used by Ha et al. [3] had only 1047 observations. Therefore, a better strategy may be to impute these MVs, i.e., to infer them from the non-missing part of the data [26]. Consequently, there is a need to explore the effect of imputation algorithms on the predictive performance of N and P.

### 1.3. Work Objectives

In addressing the limitations mentioned above, the present work aims to formalize the virtual sensing concept for nutrient monitoring. In particular, the study seeks to achieve the following objectives:To present a specification book for virtual sensor-based N and P monitoring. The specification book (i) proposes more effective baseline models, (ii) recommends performance metrics to enable adequate model comparison, (iii) recommends the different levels of predictive accuracy benchmarks based on the requirements of data quality objectives, and (iv) proposes the benchmark datasets;To assess the effect of different scalers (Standard, MinMax, MaxAbs, and Robust Scaler) and transformers (Quantile and Power Transformer) on ML algorithms and virtual sensor performance on WQ data with outliers;To assess the effect of univariate, multivariate, and nearest neighbor imputation methods in the prediction of N and P concentrations.

The remainder of this article is structured as follows. Section 2 (i) introduces the study areas and water quality data, (ii) discusses exploratory data analysis and preprocessing, and (iii) highlights some of the best practices when developing the virtual sensors while the results and discussion are presented in Section 3.

## 2. Materials and Methods

### 2.1. Description of Study Areas and Water Quality Data

The open dataset employed in this work was obtained from the UK Centre for Ecology and Hydrology’s Environmental Information Data Centre, available at https://catalogue.ceh.ac.uk/documents/db695881-eabe-416c-b128-76691b2104d8 (accessed on 17 September 2022). The hourly chemical and physical monitoring data for The Cut (a highly urbanized catchment) and the River Enborne (a relatively rural catchment) were taken between 2010 and 2012 and 2009 and 2012, respectively. Regarding land use statistics, 39% of the River Enborne catchment is designated as ‘Arable and Horticulture’ and 21% as ‘Broadleaf Woodland’ [27]. On the other hand, 30% of The Cut catchment is designated as improved grassland, 26% as arable land, and 15% as woodland [28].

These two tributaries with contrasting land-use receive sewage effluent from a small waste treatment plant (River Enborne) and major towns (The Cut) [29]. Detailed documentation of the catchments, such as maps, satellite view photos, sampling locations, and information concerning monitoring methodology and instrumentation characteristics, comes with the datasets as supporting documentation and is also provided in Wade et al. [29]. For ground-truthing the data, the two rivers were monitored using identical (i) infrastructure, (ii) chemical analysis methodologies, and (iii) monitoring sondes, and the data were validated with weekly grab sampling and laboratory analysis [29]. The parameters measured in each catchment are listed in Table 1.

As seen in Table 1, most of the parameters are common in both catchments except for nitrogen, measured as nitrate (NO_3_) in the River Enborne and ammonium (NH_4_) in The Cut. Additionally, total phosphorus was only measured in The Cut.

### 2.2. Data Analysis Frameworks

Data analysis and statistical operations were performed using the following open-source libraries (where version numbers are given in brackets) [26]:*Pandas (1.0.5)*: Used for data analysis and manipulation;*NumPy (1.18.5)*: Used for array computations;*SciPy (1.5.0)*: Used for statistical tests;*XGBoost (2.0.0)*: An optimized distributed gradient boosting library that implements ML algorithms under the gradient boosting framework;*LightGBM (3.3.2)*: A gradient boosting ML framework that uses tree-based learning algorithms;*SHAP (0.41.0)*: Used to explain individual predictions;*Scikit-Learn**(1.1.0)*: A library for ML in Python programming language;*Python (3.8.3)*: Chosen because of its growing usage in academic and industrial settings [26].

### 2.3. Exploratory Data Analysis and Preprocessing

The available data, 15,636 records for The Cut and 20,412 records for the River Enborne, are provided in Comma-Separated Values (CSV) format. Data preprocessing transforms input data into a form appropriate for modeling. In this work, exploratory analysis detected the outliers while preprocessing included data transformation, scaling, and missing values handling.

#### 2.3.1. Box and Whisker Plot Analysis: Outlier Detection

A box and whisker plot (or boxplot) was chosen for outlier detection because it provides an insightful visualization of how spread out the data are. As seen in Figure 2 (as an example), a considerable number of data points are outside the whiskers of the box plot, demeaning these outliers as normal.

#### 2.3.2. Data Transformation

To improve the predictive performance, some ML models, such as random forest and linear regression, require features to be as close as possible to normal (or Gaussian) distribution [30]. To minimize skewness and stabilize variance in variables not normally distributed, the transformation is performed by mapping the variable of interest x into the target variable y=f(x), with the common mapping function f as one of the following: the cubic root $f\left(x\right)=\sqrt[{3}]{x}$, the square root $f\left(x\right)=\sqrt{x}$, the reciprocal f(x)=1/x, the logarithm f(x)=lnx, the Yeo-Johnson [31], and Box-Cox transforms [11]. In the present work, normality was checked by computing the Fisher-Pearson coefficient of skewness using the SciPy library. Contrary to applying the same transformation based on whether the features are positively or negatively skewed, as was the case in [9], each variable was transformed using all the mentioned mapping functions, and the one that resulted in the lowest skewness was then chosen for that particular variable, as shown in Table 2.

Importantly, all these are simple transformations that do not use any learnable parameters and therefore do not risk data leakage (discussed in Section 2.4.4) when performed before data splitting. As seen in Table 2, the appropriate mapping function for flow rate in The Cut and turbidity in both catchments is the reciprocal function, although Castrillo and García [9] applied the logarithmic function. This contradiction demonstrates the importance of data analysis (or visualization) before and after each transformation.

#### 2.3.3. Data Scaling

Features such as conductivity and total reactive phosphorus in both catchments have very different scales and contain many outliers. These two characteristics can degrade the predictive performance of many ML algorithms [26]. Additionally, unscaled data can slow (or even prevent) the convergence of several gradient-based algorithms like neural networks and support vector machines. We compared the effect of MinMaxScaler, MaxAbsScaler, StandardScaler, RobustScaler, QuantileTransformer, and PowerTransformer on ML algorithms and virtual sensor performance on WQ data with outliers.

### 2.4. Virtual Sensor Development: Best Practices

The limited number of studies concerning the application of virtual sensing for nutrient monitoring indicates that the research field is still in its infancy. Therefore, this section discusses some of the best practices that will aid in interpretability and improved trust in ML models for regulatory purposes.

#### 2.4.1. Selection of Input Variables

Good practice in any ML model development is to select a subset of relevant input variables. This is to avoid overfitting and therefore improve model generalization. For virtual sensing, the requirement is also to have the minimum best subset of these predictors since it determines the surrogate sensors that will be involved in the continuous monitoring program. Industrially, most of the current real-time or near real-time monitoring applications rely on one or more of the following WQ parameters: turbidity (Turb), dissolved oxygen (DO), temperature (Temp), pH, conductivity (EC), and chlorophyll (Chl) [29,32]. Due to advances in material science and sensor technologies, it is relatively easy to measure these parameters [32]; therefore, their high-frequency monitoring provides vital WQ data required to best manage the water bodies. However, none of these parameters can identify a specific pollutant but only serve as a screening method [32]. That is, their sudden and uncharacteristic change serves as a warning against water contamination.

In the virtual sensing context, using these variables as N and P surrogates makes financial and operational sense since they are easily and economically monitored in real-time. However, this real-time monitoring is only prevalent in developed countries due to the adoption of the Water Framework Directive [33]. In less developed countries, where monitoring cost is the main barrier [13], real-time monitoring of the six predictor variables may still be expensive considering the required number of sampling points in water bodies. In such cases, the objective would be to monitor fewer variables since that would reduce the cost of purchasing and operating the surrogate sensors. A drastic deviation of the predicted N and P concentrations from baseline conditions can then be used as an early warning to trigger additional testing or initiate preventive measures.

To select the relevant predictors, the standard procedure in various water sciences modeling studies is to apply diverse variable selection methods such as best subset selection, stepwise selection with bootstrapping, partial mutual information, hierarchical partitioning, and Pearson correlation [9,12]. From the six predictors commonly measured in real-time, we sought the minimum best subset based on feature importance (or ranking) in the present work. This is crucial for interpretability since it reveals how much each predictor contributes to the observed predictions. We selected suitable surrogates based on Shapley Additive exPlanations (SHAP) feature importance. SHAP is a unified approach that explains any ML model’s output (or prediction) by connecting local explanations with coalitional game theory [34]. We selected ET as an explainer model since it is the best-performing model (discussed in Section 3). As a sanity check, we compared the chosen features with those obtained by the RF-based explainer model and observed minor differences in feature ranking. For instance, the ET-based explainer ranked nitrate predictive features (in the River Enborne) as EC, Temp, pH, DO, Turb, and Chl, while the RF-based explainer ranked them as EC, pH, Temp, Turb, Chl, and DO.

Notably, flow rate is not among the most commonly measured variables according to [29,32]. Considering that it is one of the important predictors in four of the five target variables (discussed in Section 3.1.4), it may be advisable to include it in monitoring programs, particularly in rural settings. For instance, it is the second most influential variable for TRP prediction in the River Enborne and ranks third for nitrate prediction in the same catchment based on the stepwise forward subset selection method [9]. However, it is excluded from the predictors in the present work since it is not commonly measured in real-time [29,32].

#### 2.4.2. Cross Validation

Training and testing a ML model on the same data would produce a perfect predictive score but fail dismally on yet-unseen data. This is a classic example of a situation called overfitting. To avoid it, a standard practice in any supervised ML experiment is to randomly split the data into training and testing (evaluating) sets. However, when evaluating different estimator settings (also called hyperparameters), the risk of the test set overfitting is still prevalent since the settings can be tweaked until the model performs optimally. This way, the test set information can ‘leak’ into the model resulting in the evaluation metrics no longer reporting on the generalization performance.

Therefore, yet another portion of the dataset (so-called validation set) must be held out. This is to enable the training to happen on the training set and then the evaluation on the validation set, after which the final evaluation can be performed on the test set when the experiment is deemed successful. However, partitioning data into three sets drastically reduces the number of samples used to train the mode, and the results can depend on which random pair of train and validation sets were used [26].

This problem is solved by a procedure known as cross-validation (CV). While the test set is still necessary for final evaluation, the validation set is not required when performing CV. One type of CV, known as k-fold, splits the training set into k smaller folds or sets where each one is utilized iteratively for testing the model while the remaining k − 1 folds are used as training data. The performance reported by k-fold CV is then the average of values from the loop. In the present work, the datasets were split into train and test sets using a shuffled k-fold CV with k = 10 and a 20% subset reserved for the final test.

#### 2.4.3. Hyperparameter Tuning

Hyperparameter tuning (or optimization) is a process of choosing a subset of optimal hyperparameters for ML algorithms. A Grid Search CV is currently the most widely used parameter optimization method [26]. It searches for optimal hyperparameters by exhaustively considering all parameter combinations. In the present work, a five-fold Grid Search CV was used to find the ideal hyperparameters. Increasing the hyperparameter search space is computationally very expensive; therefore, we only searched over the main parameters. For tree-based models, for example, those are the number of trees (or n_estimators) and the size of the random subsets (or max_features) in the Scikit-Learn library. The resulting optimal values did not significantly improve the models. That is, the accuracy rating levels (as proposed in Table 3) did not change. Therefore, we kept the default hyperparameters in order to speed up the training process.

#### 2.4.4. Avoiding Data Leakage

Data leakage is a common methodological pitfall in ML-based science [26]. It occurs when the model has access to the test set information before its predictive performance is evaluated, resulting in overly optimistic results with no value in practice. A common course of data leakage is not separating the train and test data subsets during all the preprocessing [26]. Although both train and test sets should receive the same transformation, the general rule (or best practice) is that these transformations should only be *learned* from the training subset, and then the test data be *transformed*, as done on the training data. A pipeline class in Scikit-Learn (implemented in this work) makes it easier to chain all the preprocessing transformations with estimators without risking data leakage.

#### 2.4.5. Imputation of Missing Values

Imputation refers to the process of replacing missing data with plausible estimates learned from non-missing data. Univariate imputation replaces the MVs in a particular feature dimension with statistical estimates such as the median, mean, or mode of non-missing values of the same feature dimension (or variable). In contrast, multivariate imputation is a more sophisticated approach that models each feature with MVs as a function of other features and utilizes that approximation for imputation [26]. On the other hand, k-nearest neighbor (kNN) imputation imputes each missing feature using available values from the nearest neighbors for the feature and averages these closest points to fill in the missing value. We assessed the effect of univariate (mean, median, and mode), multivariate (bayesian ridge, RF, extreme gradient boosting, light gradient boosting machine, and bagging regressor), and kNN imputation techniques on the predictive performance of N and P concentrations.

## 3. Results and Discussion

### 3.1. A Specification Book

To guarantee that hardware sensors can satisfy the requirements of high-quality data to meet the monitoring requirements, they undergo a rigorous certification process that tests for acceptable accuracy, precision, and resolution, among others [35]. Virtual sensing should also be held to the same standards if the model outputs are to be practically meaningful. In this context, the following requirements are crucial for a robust and rigorous virtual sensor development process:

#### 3.1.1. Standard or Benchmark Datasets

The availability of good quality data is crucial for developing ML models [36,37]. In most established machine or deep learning application areas, the standard practice is to assess the models over several freely available benchmark datasets [38]. The significance of this practice is that it (i) enables reproducible research, (ii) facilitates a more rigorous inter-model evaluation, (iii) has been the basis for some of the most significant algorithmic developments in ML research, and (iv) guarantees the model’s robustness (or precision), among other advantages [37]. On the contrary, water quality prediction studies frequently apply only one private dataset from the studied water bodies [3,12]. To our knowledge, only one study [9] used two freely available and high-quality data from two tributaries with contrasting land uses to develop their predictive model (the same data utilized in the present work). These data are relatively large, and their quality was validated by a weekly laboratory analysis [29]. Therefore, we propose these data as a benchmark, over and above the data from the case studies being considered.

#### 3.1.2. Model Performance Evaluation

Model evaluation is the core part of developing an effective ML model. Besides the need for a reliable evaluation framework for the model to be relied upon, some of the main reasons for model evaluation include [39]:Estimating the predictive and generalization performance of the model on unseen (future) data;Providing a basis for evaluating improvements in the predictive performance when adjusting (or tweaking) model parameter values;Identifying the best ML algorithm for the problem at hand by comparing different algorithms or current modeling efforts with peer-reviewed literature results.

Several studies have explored suitable evaluation metrics for traditional watershed models [40,41]. Moriasi et al. [41] synthesized the performance measures and provided performance evaluation guidelines for widely used watershed-scale models (e.g., Soil and Water Assessment Tool). Based on their synthesis, personal modeling experiences, and meta-analysis, they recommended several statistical and graphical performance measures to evaluate model performance. From the recommended performance measures, the authors [41] highlight that the most widely used are RMSE, R^2^, and NSE. The three metrics are calculated as shown in Equation (1) [9], Equation (2) [26], and Equation (3) [12]:(1)$$\mathrm{RMSE}=\sqrt{\frac{1}{n}\overset{n}{\underset{i-1}{\sum }}{\left(y_{i}-{\hat{y}}_{i}\right)}^{2}}$$
(2)$$R^{2}=1-\frac{\sum_{i-1}^{n}{\left(y_{i}-{\hat{y}}_{i}\right)}^{2}}{\sum_{i=1}^{n}{\left(y_{i}-\mu \right)}^{2}}$$
(3)$$\mathrm{NSE}=1-\frac{\sum_{i-1}^{n}{\left({\hat{y}}_{i}-y_{i}\right)}^{2}}{\sum_{i=1}^{n}{\left(y_{i}-\mu \right)}^{2}}$$
where n is the number of observations, $y_{i}$ is the true value, ${\hat{y}}_{i}$ is the predicted value, and μ is the mean of the observed values. As seen in Equations (2) and (3), the computation of R^2^ and NSE is very similar. Therefore, since R^2^ is more common in the general machine learning community [26], we recommend the consistent use of RMSE and R^2^ for developing virtual sensors.

#### 3.1.3. Accuracy Requirements

The rapid measurement of nutrient concentrations at drinking water intakes is essential for water management, especially when concentrations exceed regulatory limits. Generally, defining data quality needs and sensor specifications from various potential users (e.g., domestic, agriculture, recreation, etc.) is critical to the technology’s evolution [14]. This is due to the varying requirements of the potential users that usually lead to design trade-offs with different data specifications. For instance, commercial sensors for measuring nitrate analytes in pure water have higher accuracy, precision, and resolution than the same nitrate sensors developed for other applications. Similarly, the accuracy requirements from virtual sensors may vary based on the intended use of the predictive data. Since the accuracy requirements (or benchmarks) for virtual sensor-based nutrient monitoring are currently unavailable in the literature, we propose the state-of-the-art accuracy ratings reported in terms of the R^2^ as shown in Table 3.

The choice of these ratings is motivated by the fact that commercial sensors, as a reference, attain these accuracies based on the measured WQ variable [13], and nitrate prediction using several machine learning algorithms in the River Enborne is within these accuracies (Section 3.1.4). Although these accuracy ratings are given in terms of R^2^ alone, the performance comparison of different prediction methods should also consider the achievable RMSE. Table 3, coupled with the predictors able to predict nutrients within these accuracy levels, would form a basis for a cost–benefit analysis for monitoring program managers (further discussed in Section 3.5).

#### 3.1.4. A Baseline Model

One of the first requirements of any measurement process is that there must be a well-known and commonly accepted standard of comparison [35]. Similarly, a baseline refers to a simple and well-understood model whose performance is used as a benchmark for comparing and evaluating other models. Currently, multiple linear regression is the preferred baseline model [3,9,14]. However, WQ data are characterized by nonlinear relationships, skewed distributions, and multicollinearity [12], all of which violate the core assumptions of LR models. Thus, the field currently lacks effective baselines. That is, basic models with competitive performance for N and P prediction.

To raise the bar of baselines, we performed ‘spot checking’ on a diverse set of commonly applied linear, nonlinear, and ensemble algorithms with default hyperparameters in the Scikit-Learn library. The aim was to identify a set of algorithms that perform well in this problem domain. We evaluated decision tree (DT), random forest (RF), extreme gradient boosting (XGB), light gradient boosting machine (LGBM), multi-layer perceptron (MLP), k-nearest neighbor (kNN), support vector machine (SVM), extremely randomized trees (ET), gradient boosting regressor (GB), stochastic gradient descent (SGD), linear regression (LR), ridge regression (Ridge), and bagging regressor (BR). In case the reader is unfamiliar with any of these algorithms, we refer them to the official machine learning library [26] for details concerning the theoretical backgrounds, mathematical formulations, implementation strategies, and tips on practical use.

To evaluate the thirteen algorithms, we used Castrillo and García [9]’s experimental setting as a benchmark. That is, we (i) used the same predictors, (ii) handled the MVs by listwise deletion, and (iii) scaled the features to zero mean and unit variance. The results of nitrate spot checking, reported in terms of R^2^, are presented in Figure 3.

Although Figure 3 presents only the results of nitrate spot checking, the rating of the best performing algorithms (ET, RF, and kNN) is the same for TRP in both catchments and TP in The Cut, with RF outperforming kNN in most cases. Notably, (i) linear regression models (SGD, LR, and ridge) are the least performing algorithms in both catchments, and (ii) ET performs better than the commonly used RF for all target variables. To further test the effectiveness of ET compared to RF, we assessed the performance of ET as a function of changing input variables. We then used the RF results obtained by [9] as a basis for comparison, as shown in Table 4.

As seen in Table 4, the average improvement provided by ET is 88% for NO_3_ prediction in the Enborne, 41% for NH_4_ prediction in The Cut, and 40% for TP prediction in The Cut. The average percentage improvement for TRP prediction in both catchments is only marginal (1% in the Enborne and 3% in The Cut). Consequently, we propose ET as a baseline for future studies. Notably, ET (i) does not make linearity assumptions, (ii) is relatively straightforward conceptually, and (iii) is easily implementable in programming languages like Python, MATLAB, or R.

### 3.2. Selected Models from Spot Checking

Apart from establishing the baseline model, spot checking (Section 3.1.4) effectively identifies the best-performing models to be optimized for the problem at hand. Using the 80% predictive performance as a threshold, we selected all the models that achieved ≥80% in both catchments for further analysis. The 80% was chosen because it is the minimum proposed for virtual sensor-based nutrient monitoring, as shown in Table 3. RF, XGB, LGBM, kNN, ET, and BR are all the algorithms that fulfilled this requirement and were subsequently selected.

### 3.3. The Effect of Different Scalers

Table 5 presents nitrate (as NO_3_) prediction accuracy for the six modes with different scaling techniques in the River Enborne. RF and ET showed the highest accuracy with no scaling, while kNN showed the lowest predictive performance with 71% accuracy. However, the scaling impact on kNN is significant as the performance improves from 71% to 95% (equal to RF) with MinMax scaling. Considering that variables like conductivity and TRP have very different scales (in both catchments), this improvement is explainable since kNN is a distance-based algorithm that predicts values based on the k closest neighbors and therefore is sensitive to the scale or magnitudes of the variables. With the rest of the models, data scaling marginally improves some but also degrades the performance of others. For instance, the performance of XGB drops from 94% to 90% after Quantile transformation. This effect may be due to the fact that this transformer changes the original feature distribution and, therefore, may distort the existence of any linear correlations between variables [26]. However, this behavior is unusual since scaling is known to have no effect on tree-based models [9,12]. For instance, scaling has no effect on RF, LGBM, ET, and the decision tree-based bagging regressor, as seen in Table 5.

The performance of RF and ET with or without scaling and kNN after scaling is in the target rating, which is crucial for N monitoring in rural catchments where river water is sometimes used untreated, and infants are vulnerable to compounds like nitrates in drinking water [42].

The fact that the MinMax scaler is the best scaling technique suggests that the high number of outliers in the current datasets should indeed be considered normal since this scaler uses the minimum and maximum feature values for scaling and, therefore prone to very large marginal outliers.

Similar behavior is observed for TRP prediction in the same catchment. As seen in Table 6, the only major difference between NO_3_ and TRP is that TRP predictive performance is no longer in the target but in the acceptable rating (as defined in Table 3), although ET’s performance is relatively close to the target rating. The drop in TRP predictive performance, which was also observed for P fractions compared to NO_3_ and total nitrogen in [12], may be due to the high nonlinearity and stochastic pattern fluctuations of TRP in water bodies [43]. Additionally, the MinMax scaler is still the best scaling technique, as evidenced by the kNN performance after scaling.

In contrast to the River Enborne, where the NO_3_ predictive performance was very high, nitrate (as NH_4_) prediction in The Cut, as shown in Table 7, is below 90%, with the highest accuracy of 89% achieved by ET with or without data scaling. The second best performing model is kNN, whose accuracy improves from 21% to 88% after MinMax scaling. To explain this substantial drop in predictive performance between the two catchments, it is essential to note that the River Enborne is a rural catchment that receives sewage effluent from smaller wastewater treatment plants, while The Cut is an urban catchment dominated by treated sewage effluent from major towns and urban runoff. This implies that The Cut’s water quality dynamics are different and more complex due to various industrial activities and urban runoff [29]. The various dissolved contaminants make it difficult for ML models to detect variable relationships.

The negative impact of the Quantile transformer is more pronounced in this case. For instance, the predictive performance of XGB drastically drops from 82% to 68% after scaling. The power transformer also degrades the performance of all the models except for kNN. As mentioned already, the negative effect of these transformers may be due to the fact that they both change the original feature distribution and, hence, may distort the existence of any linear correlations between variables [26], which are already complex in urban catchments.

As with TRP prediction in the River Enborne, TRP prediction accuracy in The Cut is even lower. As shown in Table 8, only two models (RF and ET) achieve more than 80% accuracy with or without MinMax scaling. Although consistent with previous studies [12,44], this performance is in the ‘poor’ rating and thus unsatisfactory. While the reason for this poor performance is unclear, Harrison et al. [12] suggest that it may be due to the contrasting importance of particulate vs. dissolved nutrient fractions, including their respective sources. The other reason could be that TRP has three ‘extra’ electrons and can easily bond with other positively charged compounds and elements. Consequently, the formation of these ‘new’ compounds makes it difficult for ML algorithms to establish strong surrogate—regression relationships.

The effect of scaling on total phosphorus prediction in The Cut is the same as that of TRP in the same catchment. Even the predictive performances of the six models are similar. This is because TP and TRP concentrations in The Cut are strongly correlated [29]. In fact, the correlation relationship is very close to 1:1.

### 3.4. Missing Values Handling

Having established that ET is the best-performing model with or without feature scaling, we assessed the impact of various imputation methods compared to listwise deletion on its predictive performance. The results of this assessment are presented in Table 9.

For NO_3_ prediction in River Enborne, multivariate imputation with RF, BR, XGB, or LGBM as estimators produces the best results, although equal to listwise deletion. In contrast, the impact of all imputation techniques on NH_4_ prediction in The Cut is more positive. For instance, the predictive accuracy improves from the tolerable rating with listwise deletion to the acceptable rating after missing values imputation. Although this improvement is only marginal, it is crucial since the predictive performance in The Cut is, on average, unsatisfactory. For TRP prediction, multivariate imputation with RF, BR, XGB, or LGBM as estimators marginally improves the predictive performance in the River Enborne, while none of the imputation techniques is effective in The Cut. This comparison suggests that more advanced imputation techniques, such as generative adversarial imputation nets, should be explored for TRP prediction, particularly in The Cut, where the predictive performance is in the poor rating.

### 3.5. Sensitivity Analysis

Sensitivity analysis determines how the model output changes as input variables change. It enables model developers and end-users to understand the impact of each feature on the model output. Since some features can be more important (or more relevant) than others, feature importance enables their ranking based on their importance and contribution to the final model. Table 10 presents the predictive performance of the ET model on the five target variables as a function of each predictor’s contribution based on SHAP values. A table of this nature would enable a cost-benefit analysis for water managers to choose the appropriate number of real-time sensors to install based on the achievable predictive performance.

As seen in Table 10, the NO_3_ predictive performance from only three surrogates in the River Enborne is in the acceptable rating. In contrast, TRP requires at least four predictors to achieve the same rating. Notably, a set of four surrogate sensors (EC, DO, Temp, and Turb) would be able to simultaneously predict NO_3_ and TRP within the acceptable rating. Interestingly, the inclusion of Chl results in a marginal drop in performance, implying that the model considers it noise or irrelevant. In The Cut, a subset of six surrogate sensors would be able to predict NH_4_ within the acceptable rating, while TRP and TP would be within the poor rating. There is a drop in predictive performance at the introduction of the second predictor in The Cut. This suggests that at least three surrogate sensors would be necessary for this catchment. In monitoring settings where the flow rate is already measured in high-frequency, as in the River Enborne and The Cut, its inclusion as a predictor is worthwhile since it increases TRP and TP predictive performance in The Cut to more than 85%, which is the tolerable rating.

### 3.6. Performance Results of Our Approach: A Comparative Analysis

To highlight this work’s contribution, we compared the performance of our approach with the three most relevant studies [3,9,12] in the same problem domain. Firstly, we present a comparative analysis in terms of the crucial steps involved in virtual sensor development, as shown in Table 11.

Secondly, we compared the predictive performance of the three studies with ET (our work) in terms of the accuracy values reached, as shown in Table 12.

As seen in Table 12, our work outperformed the three studies in terms of the RMSE and NSE or R^2^ values reached, except for one case when the RMSE was slightly lower, although R^2^ was still superior. Since parts of the watershed studied by [12] are relatively urbanized, the TP predictive performance of 0.74 shows that the complex nature of water quality data in urban catchments does indeed make it difficult for traditional machine learning models to establish strong inferences between target variables and surrogates.

## 4. Conclusions

In the present work, an innovative ML-based virtual sensing concept for continuous monitoring of N and P is discussed. The effectiveness of the virtual sensors is verified using two catchments of the River Thames (in the UK) with contrasting land uses. Using only the predictors most commonly measured in real-time as inputs, the predictive performances of the ET-based virtual sensors were higher than XGB, LGBM, kNN, BR, and the commonly applied RF model. A summary of this work’s contributions is as follows:This study presented a specification book and best practices for developing virtual sensors for nutrient monitoring. These concepts will aid in developing robust and interpretable virtual sensors whose performance can be independently verified;The effect of various scalers and transformers on WQ data with outliers is assessed. Although this is usually overlooked in WQ monitoring studies, the present work proved its importance. For instance, the most effective scaler was the MinMax scaler and not the commonly applied standard scaler;Even though missing data is a common problem in WQ data, it is often overlooked in WQ monitoring studies. Therefore, this study assessed the impact of missing value imputation in predicting N and P concentrations where RF and LGBM-based multivariate imputers were the most effective.

Although ET performed satisfactorily in the River Enborne (achieving R^2^ values as high as 96% with only four predictors), its limitation is that the predictive performance, particularly for TRP and TP in The Cut, is still unsatisfactory. The required number of surrogate sensors (six) to achieve the acceptable accuracy rating for NH_4_ prediction is too high. Additionally, the same surrogates can only provide 82% accuracy for TRP and TP.

Notably, the 82% was achieved when missing values were handled by listwise deletion, which reduced the size of the dataset from 15,636 to 8934. Since none of the applied imputation methods resulted in better performance in this case, a realistic strategy includes the exploration of more advanced imputation and predictive techniques for developing virtual sensors in more urbanized watersheds. Therefore, future works will explore (i) the effect of data augmentation using generative adversarial networks and (ii) deep learning models such as deep echo state networks as predictive models.

## Figures and Tables

**Figure 1 sensors-22-07338-f001:**
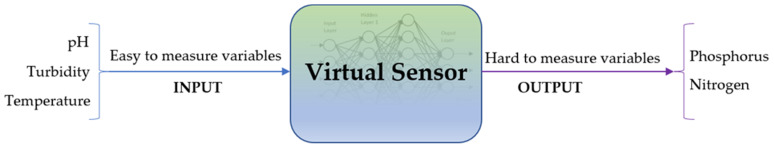
A working principle of the virtual sensing system.

**Figure 2 sensors-22-07338-f002:**
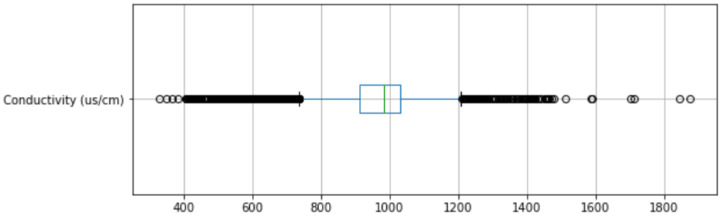
A boxplot showing conductivity outliers in The Cut.

**Figure 3 sensors-22-07338-f003:**
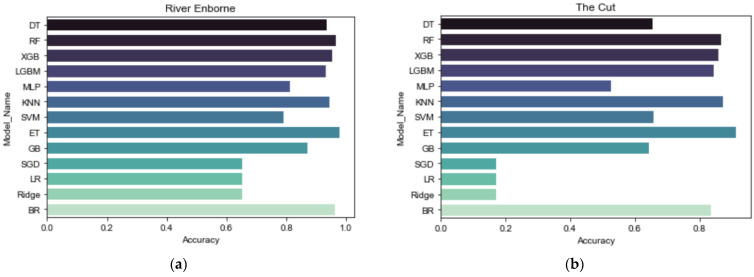
(**a**) Spot checking nitrate predictive performance in the River Enborne; (**b**) Spot checking nitrate predictive performance in The Cut.

**Table 1 sensors-22-07338-t001:** Datasets used for virtual sensor development: parameters measured in each catchment, their chemical formulas, and descriptions where N/A denotes the absence of standard formulas and • indicates the catchment where the variable was measured.

Predictors	Formula	The Cut	Enborne	Description
pH	N/A	•	•	A measure of water’s acidity or basicity. A changing stream pH indicates an increase in water pollution.
Flow rate	N/A	•	•	The volume of water flowing past a point per unit time. Streamflow and runoff drive the generation and delivery of various diffuse (non-point) pollutants; therefore, the knowledge of streamflow enables the determination of pollutant loads.
Turbidity	N/A	•	•	A measure of water’s relative clarity. It is an optical characteristic that indicates the presence of bacteria, pathogens, and other harmful contaminants.
Chlorophyll	C_55_H_72_O_5_N_4_Mg	•	•	A measure of how much algae is growing in a water body. It is usually used to classify the water body’s trophic condition.
Temperature	N/A	•	•	A property that expresses how cold or hot the water is. It influences various other variables and can alter water’s chemical and physical properties.
Conductivity	N/A	•	•	A measure of water’s ability to conduct electricity. Its increase may indicate that a discharge has decreased the water body’s relative health or condition.
Dissolved oxygen	O_2_	•	•	A measure of how much non-compound oxygen is available in the water. It is a direct indicator of the water body’s ability to support aquatic life.
Nitrogen as nitrate	NO_3_		•	Nitrates are one form of nitrogen found in aquatic environments. Although nitrates are vital plant nutrients, their excess amounts can accelerate eutrophication.
Nitrogen as ammonium	NH_4_	•		Ammonium is another form of nitrogen found in water bodies. It has toxic effects on aquatic life at elevated concentrations.
Total phosphorus	P	•		Total phosphorus is more stable and, therefore, a more reliable index of the phosphorus status in water bodies. Similar to nitrates, excess amounts of phosphorus lead to eutrophication and harmful algal growth.
Total reactive phosphorus	PO_4_^3−^	•	•	Total reactive phosphorus (orthophosphate) is regarded as the best indicator of the nutrient status of water bodies. It has similar effects to nitrates and ammonium in excess amounts.

**Table 2 sensors-22-07338-t002:** Transformation of variables in each catchment.

Variable	Transformation	
	The Cut	River Enborne
Flow rate (Flow)	Reciprocal	Logarithm
Chlorophyll (Chl)	Logarithm	Logarithm
Dissolved oxygen (DO)	Square root	Logarithm
Nitrate (as NH_4_ or NO_3_)	Cube root	Cube root
Turbidity (Turb)	Reciprocal	Reciprocal
Total Reactive Phosphorus (TRP)	None	Cube root
pH	None	Reciprocal
Conductivity (EC)	None	None
Temperature (Temp)	None	None
Total Phosphorus (TP)	Square root	

**Table 4 sensors-22-07338-t004:** Performance comparison of RF and ET models using RMSE ± standard deviation (std).

Predictors in RF and ET Models	RF: [9]	ET: Our Work	Improvement (%)
	RMSE ± Std	RMSE ± Std	
**NO_3_ in River Enborne**			
EC	0.458 ± 0.286	0.062 ± 0.002	86%
EC, pH	0.343 ± 0.229	0.050 ± 0.002	85%
EC, pH, Flow	0.254 ± 0.186	0.027 ± 0.001	89%
EC, pH, Flow, Temp	0.194 ± 0.138	0.017 ± 0.001	91%
**TRP in River Enborne**			
EC	0.061 ± 0.043	0.066 ± 0.002	−8%
EC, Flow	0.043 ± 0.035	0.051 ± 0.001	−19%
EC, Flow, Temp	0.030 ± 0.025	0.027 ± 0.001	10%
EC, Flow, Temp, Turb	0.025 ± 0.021	0.020 ± 0.001	20%
**NH_4_ in The Cut**			
Chl	0.210 ± 0.144	0.130 ± 0.004	38%
Chl, Temp	0.190 ± 0.123	0.135 ± 0.003	29%
Chl, Temp, Turb	0.150 ± 0.104	0.091 ± 0.004	39%
Chl, Temp, Turb, pH	0.128 ± 0.095	0.067 ± 0.004	48%
Chl, Temp, Turb, pH, EC	0.107 ± 0.077	0.051 ± 0.003	52%
**TRP in The Cut**			
EC	0.199 ± 0.115	0.196 ± 0.005	2%
EC, Turb	0.180 ± 0.112	0.205 ± 0.007	−14%
EC, Turb, Temp	0.141 ± 0.099	0.142 ± 0.003	−1%
EC, Turb, Temp, pH	0.117 ± 0.088	0.107 ± 0.005	9%
EC, Turb, Temp, pH, Flow	0.107 ± 0.078	0.088 ± 0.004	18%
**TP in The Cut**			
EC	0.192 ± 0.114	0.122 ± 0.003	36%
EC, Turb	0.173 ± 0.116	0.128 ± 0.004	26%
EC, Turb, Temp	0.153 ± 0.104	0.088 ± 0.002	42%
EC, Turb, Temp, pH	0.119 ± 0.087	0.067 ± 0.003	44%
EC, Turb, Temp, pH, Flow	0.110 ± 0.081	0.055 ± 0.002	50%

**Table 5 sensors-22-07338-t005:** The effect of different scalers on NO_3_ prediction in the River Enborne.

Scaling Method	RF		XGB		LGBM		kNN		ET		BR	
	RMSE	R^2^	RMSE	R^2^	RMSE	R^2^	RMSE	R^2^	RMSE	R^2^	RMSE	R^2^
No scaling	0.0211	0.9544	0.0246	0.9382	0.0279	0.9200	0.0534	0.7085	0.0176	0.9682	0.0229	0.9465
Robust scaler	0.0211	0.9547	0.0241	0.9405	0.0280	0.9198	0.0257	0.9322	0.0176	0.9684	0.0226	0.9469
MaxAbs scaler	0.0211	0.9546	0.0246	0.9378	0.0279	0.9200	0.0292	0.9125	0.0176	0.9681	0.0224	0.9466
MinMax scaler	0.0211	0.9548	0.0241	0.9403	0.0279	0.9200	0.0210	0.9548	0.0176	0.9683	0.0226	0.9467
Standard scaler	0.0210	0.9546	0.0243	0.9397	0.0278	0.9207	0.0236	0.9427	0.0176	0.9683	0.0226	0.9459
Power transformer	0.0211	0.9545	0.0242	0.9398	0.0279	0.9200	0.0234	0.9440	0.0176	0.9684	0.0227	0.9494
Quantile transformer	0.0223	0.9487	0.0313	0.8999	0.0315	0.8984	0.0253	0.9344	0.0199	0.9596	0.0236	0.9429

**Table 6 sensors-22-07338-t006:** The effect of different scalers on TRP prediction in the River Enborne.

Scaling Method	RF		XGB		LGBM		kNN		ET		BR	
	RMSE	R^2^	RMSE	R^2^	RMSE	R^2^	RMSE	R^2^	RMSE	R^2^	RMSE	R^2^
No scaling	0.0268	0.9386	0.0298	0.9240	0.0322	0.9112	0.0565	0.7268	0.0243	0.9498	0.0284	0.9318
Robust scaler	0.0269	0.9377	0.0299	0.9234	0.0323	0.9108	0.0311	0.9170	0.0243	0.9497	0.0285	0.9282
MaxAbs scaler	0.0268	0.9383	0.0297	0.9244	0.0322	0.9112	0.0331	0.9065	0.0243	0.9495	0.0283	0.9295
MinMax scaler	0.0269	0.9381	0.0299	0.9234	0.0324	0.9103	0.0270	0.9377	0.0242	0.9498	0.0287	0.9302
Standard scaler	0.0268	0.9379	0.0298	0.9242	0.0325	0.9099	0.0294	0.9260	0.0241	0.9498	0.0290	0.9299
Power transformer	0.0269	0.9385	0.0299	0.9237	0.0324	0.9102	0.0294	0.9258	0.0241	0.9490	0.0287	0.9300
Quantile transformer	0.0293	0.9269	0.0354	0.8926	0.0360	0.8892	0.0316	0.9145	0.0276	0.9343	0.0302	0.9222

**Table 7 sensors-22-07338-t007:** The effect of different scalers on NH_4_ prediction in The Cut.

Scaling Method	RF		XGB		LGBM		kNN		ET		BR	
	RMSE	R^2^	RMSE	R^2^	RMSE	R^2^	RMSE	R^2^	RMSE	R^2^	RMSE	R^2^
No scaling	0.0494	0.8552	0.0554	0.8183	0.0560	0.8140	0.1157	0.2085	0.0436	0.8856	0.0538	0.8299
Robust scaler	0.0494	0.8540	0.0548	0.8218	0.0558	0.8157	0.0501	0.8508	0.0438	0.8852	0.0529	0.8275
MaxAbs scaler	0.0497	0.8538	0.0554	0.8183	0.0560	0.8140	0.0536	0.8287	0.0440	0.8850	0.0543	0.8309
MinMax scaler	0.0495	0.8520	0.0540	0.8275	0.0559	0.8151	0.0445	0.8821	0.0438	0.8859	0.0537	0.8298
Standard scaler	0.0496	0.8544	0.0550	0.8208	0.0560	0.8141	0.0463	0.8726	0.0440	0.8842	0.0524	0.8312
Power transformer	0.0567	0.8093	0.0577	0.8024	0.0590	0.7934	0.0549	0.8183	0.0514	0.8385	0.0599	0.7903
Quantile transformer	0.0631	0.7611	0.0732	0.6832	0.0767	0.6524	0.0562	0.8118	0.0627	0.7678	0.0642	0.7670

**Table 8 sensors-22-07338-t008:** The effect of different scalers on TRP prediction in The Cut.

Scaling Method	RF		XGB		LGBM		kNN		ET		BR	
	RMSE	R^2^	RMSE	R^2^	RMSE	R^2^	RMSE	R^2^	RMSE	R^2^	RMSE	R^2^
No scaling	0.0954	0.8058	0.1040	0.7675	0.1079	0.7498	0.1723	0.3623	0.0904	0.8229	0.1022	0.7805
Robust scaler	0.0950	0.8059	0.1042	0.7667	0.1080	0.7496	0.1044	0.7660	0.0909	0.8227	0.1019	0.7784
MaxAbs scaler	0.0956	0.8047	0.1045	0.7651	0.1079	0.7498	0.1085	0.7471	0.0904	0.8229	0.1019	0.7782
MinMax scaler	0.0951	0.8041	0.1044	0.7657	0.1079	0.7499	0.0983	0.7926	0.0906	0.8221	0.1009	0.7795
Standard scaler	0.0952	0.8041	0.1037	0.7687	0.1082	0.7486	0.1000	0.7849	0.0909	0.8222	0.1014	0.7798
Power transformer	0.1017	0.7792	0.1054	0.7612	0.1080	0.7493	0.1069	0.7538	0.0974	0.7951	0.1090	0.7529
Quantile transformer	0.1006	0.7820	0.1110	0.7353	0.1133	0.7242	0.1038	0.7684	0.1005	0.7837	0.1051	0.7564

**Table 9 sensors-22-07338-t009:** Performance comparison of the ET model using different imputation methods.

Type	Method(s)	River Enborne				The Cut			
		NO_3_		TRP		NH_4_		TRP	
		RMSE	R^2^	RMSE	R^2^	RMSE	R^2^	RMSE	R^2^
Deletion	Listwise	0.0176	0.9683	0.0242	0.9498	0.0438	0.8859	0.0906	0.8221
Univariate	Mean	0.0282	0.9018	0.0335	0.8624	0.0427	0.9053	0.0783	0.7794
	Mode	0.0285	0.9004	0.0461	0.8141	0.0426	0.9054	0.0781	0.7853
	Median	0.0282	0.9010	0.0337	0.8629	0.0424	0.9057	0.0780	0.7775
Multivariate	Bayesian ridge	0.0215	0.9463	0.0245	0.9370	0.0430	0.9033	0.0760	0.7957
	RF	0.0176	0.9649	0.0208	0.9580	0.0429	0.9058	0.0798	0.7948
	BR	0.0178	0.9635	0.0217	0.9542	0.0427	0.9051	0.0819	0.7893
	XGB	0.0183	0.9687	0.0217	0.9548	0.0428	0.9040	0.0794	0.7904
	LGBM	0.0181	0.9678	0.0212	0.9555	0.0426	0.9098	0.0763	0.8009
Nearest neighbors	kNN	0.0238	0.9328	0.0320	0.8884	0.0436	0.9017	0.0853	0.7949

**Table 10 sensors-22-07338-t010:** Predictive performance of the ET model as a function of each predictor’s contribution.

Predictors in the ET Model	RMSE	R^2^
**NO_3_ in River Enborne**		
EC	0.0617	0.6107
EC, Temp	0.0559	0.6818
EC, Temp, pH	0.0274	0.9223
EC, Temp, pH, DO	0.0205	0.9566
EC, Temp, pH, DO, Turb	0.0172	0.9695
EC, Temp, pH, DO, Turb, Chl	0.0177	0.9681
**TRP in River Enborne**		
EC	0.0666	0.5637
EC, DO	0.0608	0.6355
EC, DO, Temp	0.0343	0.8848
EC, DO, Temp, Turb	0.0257	0.9345
EC, DO, Temp, Turb, pH	0.0213	0.9559
EC, DO, Temp, Turb, pH, Chl	0.0212	0.9558
**NH_4_ in The Cut**		
Temp	0.1312	0.1620
Temp, Chl	0.1342	0.1220
Temp, Chl, Turb	0.0907	0.5986
Temp, Chl, Turb, EC	0.0655	0.7895
Temp, Chl, Turb, EC, DO	0.0526	0.8647
Temp, Chl, Turb, EC, DO, pH	0.0429	0.9101
**TRP in The Cut**		
EC	0.1952	0.1820
EC, Turb	0.2037	0.1072
EC, Turb, DO	0.1554	0.4813
EC, Turb, DO, Temp	0.1101	0.7401
EC, Turb, DO, Temp, Chl	0.0999	0.7864
EC, Turb, DO, Temp, Chl, pH	0.0907	0.8219
**TP in The Cut**		
EC	0.1213	0.1697
EC, DO	0.1291	0.0593
EC, DO, Turb	0.0956	0.4853
EC, DO, Turb, Temp	0.0680	0.7382
EC, DO, Turb, Temp, Chl	0.0610	0.7880
EC, DO, Turb, Temp, Chl, pH	0.0556	0.8253

**Table 11 sensors-22-07338-t011:** A methodological comparison of our work with the three most related studies, where N/S = not specified, ML = machine learning, kNN = k-nearest neighbor, RF = random forest, MLR = multiple linear regression, DT = decision tree, XGB = extreme gradient boosting, LGBM = light gradient boosting machine, BR = bagging regressor, MLP = multi-layer perceptron, SVM = support vector machine, ET = extremely randomized trees, GB = gradient boosting, SGD = stochastic gradient descent, and SHAP = Shapley additive explanations.

Step	[3]	[9]	[12]	Our Work	Remark
Data transformation	N/S	Cubic Logarithm	N/S	Cubic Logarithm Reciprocal Square root	Even though water quality data is usually skewed, only [9] performed feature transformation, although sub-optimally on three features.
Data scaling	N/S	Standard scaler	N/S	Standard scaler MinMax scaler MaxAbs scaler Robust scaler Power transformer Quantile transformer	Only one study [9] scaled the data before evaluating the MLR model. However, MinMax scaler is the best scaling technique in this case. While the other two studies did not scale the data, the impact on their model (RF) would have been minimal.
Missing values handling	N/S	Listwise deletion	Median imputation	Listwise deletion kNN imputation Univariate imputation Multivariate imputation	Amongst the various missing data handling methods, our work showed that multivariate imputation, which was not implemented in the three reference studies, results in best-performing models *on average*.
ML models	RF, MLR	RF, MLR	RF	RF, DT, XGB, LGBM, BR, MLP, kNN, SVM, ET, GB, SGD, MLR, Ridge	Although the commonly used RF performed competitively, the extensive analysis in our work showed that ET performs better.
Input variable selection	N/S	Stepwise selection	Stepwise selection	SHAP	Contrary to the commonly used stepwise selection method, we applied SHAP in this work because it satisfies the interpretability requirements, which are essential for sensitive applications like public health.

**Table 12 sensors-22-07338-t012:** A comparative analysis (in terms of the accuracy values reached) of our work and the three most related studies. The predictive performance is not reported where there is a hyphen.

Method	NO_3_		TRP		TP	
	RMSE	NSE/R^2^	RMSE	NSE/R^2^	RMSE	NSE/R^2^
Random forest [3]	0.059	0.89	**0.005**	0.903	-	-
Random forest [9]	0.194	-	0.025	-	0.110	-
Random forest [12]	0.120	0.89	2.000	0.320	11.00	0.740
Extra trees [our work]	**0.017**	**0.97**	0.021	**0.956**	**0.056**	**0.825**

**Table 3 sensors-22-07338-t003:** Proposed accuracy requirements for virtual sensor-based nutrient monitoring.

Accuracy Metric	Accuracy Ratings			
	Target	Acceptable	Tolerable	Poor
R^2^	95–100%	90–94%	85–89%	80–84%

## Data Availability

Publicly available datasets were analyzed in this study. This data can be found here: https://catalogue.ceh.ac.uk/documents/db695881-eabe-416c-b128-76691b2104d8, accessed on 17 September 2022.

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
