# Peer review of "A Virtual Sensing Concept for Nitrogen and Phosphorus Monitoring Using Machine Learning Techniques"

_sensors, 2022, doi:10.3390/s22197338_

Round 1

Reviewer 1 Report

The manuscript “A Virtual Sensing Concept for Nitrogen and Phosphorus Monitoring Using Machine Learning Techniques” uses an innovative ML-based virtual sensing concept for continuous monitoring of N and P. It is a nice try to predict difficulty and discontinuously variables (e.g., N and P) by using other easily and continuously measurable water quality variables. This manuscript uses random forest, extremely randomized trees (ET), extreme gradient boosting, k-nearest neighbors, light gradient boosting machine, and bagging regressor-based virtual sensors to predict N and P in two catchments with contrasting land uses. The effect of data scaling and missing value imputation were also assessed, while the Shapley additive explanations were used to rank feature importance. A specification book, sensitivity analysis, and best practices for developing virtual sensors are discussed, too. However, there are some issues that need to be further clarified before being published.

 1.      Better to provide the statistics of land use in two catchments of the River Thames. So the audience could know how effective the virtual sensor is.

2.      The authors select the flow rate for their input variables. Since it is one of the important predictors in four of the five target variablesTable 4). It is recommended to further explain its effectiveness on N and P.

3.      Better to reorganized the text. The Sections Materials and Methods, Introduction and Results and Discussion cannot be clearly distinguished.

4.      Methodological comparisons of related studies should be performed with the same number of input variables.

5.      For predictor variables, only the TRP parameters are common in both catchments. So it is better to supplement more data.

6.      This manuscript is more like a methodological report than a research article.

Reviewer 2 Report

The manuscript deals with the alternative how to overcome problems of instantaneous and intermittent monitoring of N and P in the environment. The idea is to present a virtual sensor to overcome typical problems faced with during in situ measurements. 

No significant need for improvements has been detected. Manuscript presents a complete and clear presentation of the virtual sensor development, its application and effectiveness. 

Congratulations to the authors.

Reviewer 3 Report

Hello,

the publication is very interesting, laid out clearly. The topic is very broad, a lot of data.

However, a few observations:

1. Maps of where water samples were taken are missing. The water intake scheme is missing.

2. Ionic chemical formulas must be written in Table 1 and its explanation.

3. 2.2. part of the Data Analysis Framework is too laconic. It must be explained in more detail which indicators were evaluated by which program.

4. Lack of discussion.

5. Conclusions are too broad, descriptive in nature. You need to specify. They have to answer the set tasks.

Round 2

Reviewer 1 Report

The manuscript has been much improved after revision.